# Evaluation of choroidal thickness in prodromal Alzheimer's disease defined by amyloid PET

Alicia López-de-Eguileta[1]*, Carmen Lage[2], Sara López-García[2], Ana Pozueta[2],
María García-Martínez[2], Martha Kazimierczak[2], María Bravo[2], María de Arcocha-Torres[3],
Ignacio Banzo[3], Julio Jimenez-Bonilla[3], Andrea Cerveró[1], Alexander Goikoetxea[4],
Eloy Rodríguez-Rodríguez[2], Pascual Sánchez-Juan[2], Alfonso Casado[1]*

1 Department of Ophthalmology, 'Marqués de Valdecilla' University Hospital, University of Cantabria,
Institute for Research 'Marqués de Valdecilla', Santander, Spain, 2 Neurology Department and Centro de
Investigación Biomédica en Red sobre Enfermedades Neurodegenerativas, 'Marqués de Valdecilla'
University Hospital, University of Cantabria, Institute for Research 'Marqués de Valdecilla', Santander, Spain,
3 Nuclear Medicine Department, University Hospital Marqués de Valdecilla, University of Cantabria,
Molecular Imaging Group—IDIVAL, Santander, Spain, 4 Department of Anatomy, University of Otago,
Dunedin, New Zealand

* alicialeguileta@gmail.com (ALE); casadorojo@hotmail.es (AC)

## Abstract

### Objective

To assess and compare the involvement of choroidal thickness (CT) in patients with mild cognitive impairment (MCI) and dementia due to Alzheimer's disease (AD) defined by amyloid PET and healthy controls (HC).

### Methods

Sixty-three eyes from 34 AD patients [12 eyes (19.0%) with dementia and 51 eyes (80.9%) with MCI], positive to [11]C-labelled Pittsburgh Compound-B with positron emission tomography ([11]C-PiB PET/CT), and the same number of sex- and age-paired HC were recruited. All participants underwent enhanced depth imaging optical coherence tomography (EDI-OCT) assessing CT at 14 measurements from 2 B-scans. Paired Student t-test was used to compare CT measurements between MCI, dementia and sex- and age-paired HC. A univariate generalized estimating equations model (GEE) test was performed to compare MCI and dementia individually with all HC included.

### Results

Compared with HC, eyes from patients with positive [11]C-PiB PET/CT showed a significant CT thinning in 5 selected locations (in foveal thickness in vertical scan, in temporal scan at 1500μm, in superior scan at 500μm and in inferior scan at 1000μm and 1500μm, p = 0.020–0.045) whilst few significant CT reduction data was reported in MCI or dementia individually versus HC. However, the GEE test identified significant CT thinning in AD compared with all HC included (p = 0.015–0.046).

**Data Availability Statement:** All relevant data are within the manuscript and its Supporting Information files.

**Funding:** Dr. Sánchez-Juan was supported by grants from IDIVAL, Instituto de Salud Carlos III (Fondo de Investigación Sanitario, PI08/0139, PI12/02288, PI16/01652, JPND (DEMTEST PI11/03028) and the CIBERNED program and Siemens Healthineers (Valdecilla Cohort for Memory and Brain Aging). The funders had no role in study design, data collection and analysis, decision to publish, or preparation of the manuscript.

**Competing interests:** Dr. Sánchez-Juan received funding from Siemens Healthineers. This does not alter our adherence to PLOS ONE policies on sharing data and materials.

## Conclusions

To our knowledge, the present study is the first measuring CT in eyes from MCI and dementia eyes positive to $^{11}$C-PiB PET/CT reporting a significant trend towards CT thinning in MCI patients which became more pronounced in dementia stage. We support further investigation involving larger and prospective OCT studies in AD population characterized with available biomarkers to describe whether choroidal vascular damage occurs specifically in prodromal stages of AD.

## Introduction

Alzheimer's disease (AD) is a neurodegenerative disorder and is the most common cause of dementia and one of the leading sources of morbidity and mortality in the aging population [1]. Globally, an estimated 47 million people are affected by dementia and the incidence doubles every 10 years after age 60 years approximately [2].

The hallmark neuropathologic changes of AD are extracellular beta amyloid beta plaques and neurofibrillary tangles (NFT) comprised of intracellular hyperphosphorylated tau protein (p-tau). These neuropathological changes are believed to start 15–20 years before the onset of clinical symptoms of dementia [3]. A definitive diagnosis of AD requires histopathologic post-mortem examination.

Clinical criteria for the diagnosis of AD have evolved over time and current criteria have been established by the National Institute on Aging and the Alzheimer's Association (NIA-AA) updated in 2011 [4, 5]. The ability to accurately diagnosis AD has improved with the emergence of new laboratory biomarkers and imaging techniques to measure such neuropathologic damage in vivo [6]. Aβ protein brain deposition is detected by decreased 42–amino acid form of Aβ (Aβ-42) levels in cerebrospinal fluid (CSF) and positron emission tomography/computed tomography (PET/CT) imaging using $^{11}$C-labeled Pittsburgh Compound-B ($^{11}$C-PiB) ligand [3, 7], which is the most studied and validated PET marker of Aβ. A proper application of $^{11}$C-PiB PET/CT would be useful to predict the conversion of MCI to AD. The sensitivity and specificity of $^{11}$C-PIB-PET for predicting conversion to AD ranged from 83.3% to 100% and 41.1% to 100%, respectively [8]. Besides, biomarkers of tau deposition (a component of NFT) include increased CSF total tau (T-tau) and phosphorylated tau (P-tau). In addition to the molecular biomarkers, there are several topographic biomarkers used to assess brain changes that correlate with the regional distribution of neuronal dysfunction associated with AD [9]. Neurodegeneration is related to cortical atrophy on magnetic resonance imaging (MRI) and hypometabolism on fluorodeoxyglucose-PET/CT (FDG-PET/CT) [10]. In fact, AD biomarkers have shown many potential clinical benefits, such as preclinical detection of AD and an accurate differentiation of AD from dementias of other etiologies [11, 12]. However, restrictions still exist in clinical practice (such as standardization problems and invasiveness in the case of CSF markers, and high costs and limited availability in the case of amyloid PET) and they are not yet recommended for routine diagnostic purposes [10, 13].

Nowadays, research is focused on the diagnosis of AD at early stages in an effort to define properly prodromal and preclinical forms of AD for design early-intervention clinical trials in order to apply potential treatments before the damage is established. Mild cognitive impairment (MCI) is an intermediate stage between normal aging and early dementia characterized by cognitive deficits primarily affecting memory with preserved overall cognitive and

functional abilities and the absence of a dementia [3, 14]. The specific designation of MCI due to AD is used when a biomarker associated with AD is present [15].

Investigation of new biomarkers has involved the evaluation of the eye, as AD pathogenesis is associated with impairments in visual function [16]. Several evidence indicates that AD also affects the retina, a developmental outgrowth of the brain [17–19], possibly causing these symptoms. Among the characteristics it shares with the brain, the retina contains neurons, astroglia, microglia, microvasculature with similar morphological and physiological properties [17–20].

Optical coherence tomography (OCT) is a non-invasive imaging device used clinically to evaluate a variety of ophthalmic and systemic diseases, as glaucoma or multiple sclerosis [21, 22]. Hence, several reports demonstrated retinal nerve fiber layer (RNFL) thinning [23, 24], retinal ganglion cell layer (RGCL) degeneration [24, 25] and choroidal thinning [26–30] in patients with dementia or MCI due to AD. The choroid is a vascular structure lying under retinal pigmentary epithelium (RPE) and is regulated by the autonomic nervous system. The choroidal blood nourishes the outer layers of the retina (photoreceptors) and the RPE which maintains the outer blood-retinal barrier [31]. Several researchers observed A$\beta$ deposits in choroidal vascular tissue in a mouse model of AD and in post-mortem analysis of the eyes from AD patients. Based on these findings, they proposed that accumulation of A$\beta$ in the choroid may cause vascular damage in accordance with the development of angiopathy in the brain due to A$\beta$ deposits [32, 33]. It is currently possible to investigate *in vivo* the involvement of CT in AD. It could be assessed using spectral-domain OCT (SD-OCT), with the enhanced depth imaging modality (EDI) technology [34]. Choroidal thinning was reported in dementia patients [26–29] and in MCI patients [30] through spectral-domain OCT (SD-OCT) using EDI technology. However, these previous studies suffered some limitations. Firstly, they only used neuropsychological tests (mainly MMSE) for AD diagnosis among their inclusion criteria. Besides, all these studies were performed in patients with dementia due to AD, except one of them which included MCI patients [30].

For this reason, we conducted a study to assess anatomical variations in the CT in patients with MCI and dementia due to AD defined by positive [11]C-PiB PET/CT, to determine whether CT is reduced compared with control subjects.

## Methods

### Patient/subject groups

We conducted a cross-sectional study including patients in the AD continuum (MCI and dementia) with positive [11]C-PiB PET/CT (MCI and AD) compared with cognitively healthy age- and gender-matched controls recruited consecutively from the Neurology and Ophthalmology departments of the University Hospital Marqués de Valdecilla (UHMV), between May 2016 and June 2018. The study cohort of the present submission overlaps with our previous work [25].

Healthy control subjects (HC) were volunteers recruited among family members of patients attending the ophthalmology clinic with a complaint of dry eye.

The study protocol and the written consent was approved by the Ethics Committee of the UHMV, and it was performed in accordance with the principles of the Declaration of Helsinki. Written consent forms were signed by all participants prior to examinations. All patients enrolled were able to understand the information contained in the written consent and they were not legally incompetent.

**Inclusion and exclusion criteria.** All were outpatients that met research diagnostic criteria for probable AD MCI or AD dementia with evidence of the AD pathophysiological process

(in our case defined by a positive amyloid-PET) following the recommendations of the National Institute on Aging-Alzheimer's Association [35]. Clinical diagnoses as MCI or dementia were established by a committee of four neurologists (SLG, PSJ, ERR, and CL). The differentiation of dementia from MCI rests on the determination of whether or not there is significant interference in the ability to function at work or in usual daily activities.

All patients were assessed to exclude other neurological or psychiatric etiologies and they underwent a comprehensive neuropsychological battery conducted by two trained neuropsychologists (AP, MGM), that included the main cognitive domains (memory, language, praxis, visual perception, and frontal functions). Besides, all patients underwent [11]C-PiB PET/CT at the Nuclear Medicine Department of the UHMV. [11]C-PiB synthesis and image acquisition have been described elsewhere [36]. PET/CT scans were visually interpreted by two experienced nuclear medicine and radiology specialists (JJB, IB) as positive or negative for cortical PiB uptake (Fig 1).

All participants underwent a thorough ophthalmic examination on the day of OCT imaging, by order of eye assessments: best-corrected visual acuity (Snellen charts), anterior segment biomicroscopy, refraction, OCT measurements, axial length (AL) assessment, IOP quantification with Goldmann applanation tonometer (GAT) and dilated fundus examination. Participants received one drop of tropicamide 1% and phenylephrine per eye for pupil dilation after IOP measurement to avoid modifications in choroidal thickness due to phenylephrine instillation as it has been previously reported [37]. The refractive error was recorded using an auto refractometer Canon RK-F1 (Canon USA Inc., Lake Success, NY, USA). Axial length (AL) was measured by Lenstar LS 900 (Haag Streit AG, Koeniz, Switzerland). Each individual was randomised to decide which eye was to be examined first, using the method described by Dulku [38].

Exclusion criteria included a refractive error $> 6.0$ or $< 6.0$ diopters (D) of spherical equivalent or 3.0 D of astigmatism, any history or showing evidence of ocular surgery, ocular disease such as central serous chorioretinopathy, pachychoroid spectrum, uveitis and related macular degeneration, best corrected visual acuity as poor as 20/40, intraocular pressure (IOP) $\geq 18$ mmHg, past history of raised IOP, neuroretinal rim notching, or optic disc hemorrhages. Similarly, other exclusion criteria included clinically relevant opacities of the optic media and

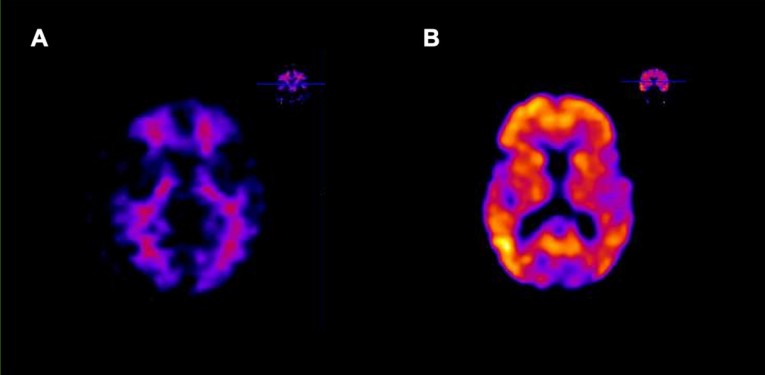

**Fig 1. Positron emission tomography/computed tomography imaging using [11]C-labeled Pittsburgh Compound-B ligand.** Positron emission tomography/computed tomography (PET/CT) imaging tracers provide a quantitative in vivo measure of the insoluble cortical beta amyloid (A$\beta$) load. [11]C-Pittsburgh compound B ([11]C-PIB) shows a nanomolar affinity for the extracellular and intravascular fibrillar deposits of A$\beta$ and a low affinity toward the amorphous amyloid deposits, soluble A$\beta$, and intracellular NFTs. Negative amyloid-PET (A) and positive amyloid-PET (B).

low-quality images due to unstable fixation, or severe cataract. Patients with mild to moderate cataract might be enrolled in the study, but only high-quality images were included. All acquired spectral domain-OCT data sets had a quality score(Q)>25. Subjects with a history of neurological or psychiatric disorder, any significant systemic illness or other serious chronic systemic diseases such as diabetes, nephrological diseases and hemodialysis, poor collaboration due to neurological dementia stage (some patients could not complete the evaluation because they became fatigued or they were not able to follow the instructions) or unstable medical condition (e.g., active cardiovascular disease), and current use of any medications known to affect cognition (e.g. sedative narcotics) were also excluded.

### Optical coherence tomography assessment

OCT measurements were taken using Spectralis OCT (Heidelberg Engineering, Dossenheim, Germany). OCT examinations were performed by an ophthalmologist (AC), who was blinded to neurological status. The examinations included one horizontal and vertical non-isotropic scans, that measures 8741 μm, resulting in 8741 x 8741 μm$^2$ dimensions. CT measurements were taken between 5 and 6 p.m. hours in all subjects [39]. Participants were asked not to consume caffeine for at least 12h before examination.

CT was measured by two raters (AL, AC). The border of CT was defined as extending from the outer portion of the hyperreflective line (corresponding to the RPE) to the inner surface of the sclera. CT was measured at 14 different locations (Fig 2): at the fovea (with horizontal and vertical scan: $F_H$ and $F_V$, respectively), and at 500, 1000 and 1500 μm from the fovea in the nasal ($N_{500μm}$, $N_{1000μm}$ and $N_{1500μm}$, respectively), temporal ($T_{500μm}$, $T_{1000μm}$ and $T_{1500μm}$, respectively), superior ($S_{500μm}$, $S_{1000μm}$ and $S_{1500μm}$, respectively), and inferior ($I_{500μm}$, $I_{1000μm}$ and $I_{1500μm}$, respectively) quadrants, as published previously [37].

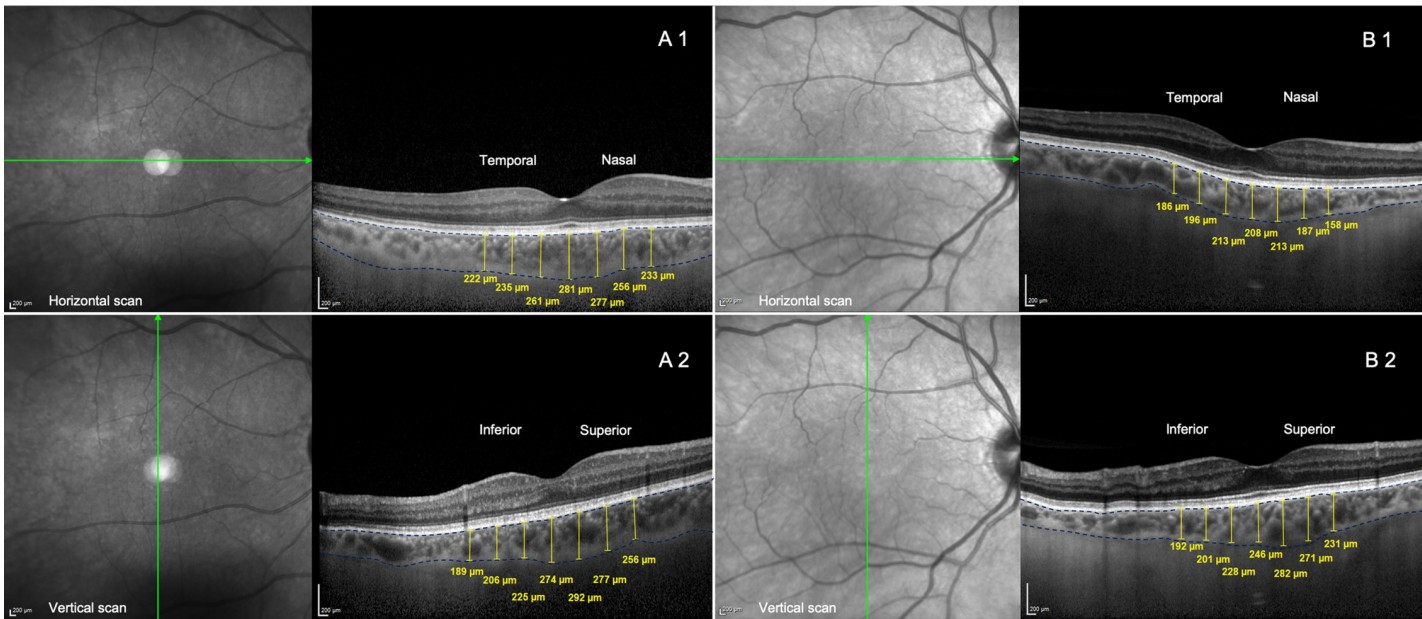

**Fig 2. Choroidal thickness measurements by EDI-OCT.** A representation of choroidal thickness (CT) measurements in the right eye that could be depicted in a patient with Alzheimer disease (A) and in the sex- and age-paired control eye (B) by optical coherence tomography. (1) CT measurements performed in the horizontal scan: subfoveal and at 500 μm, 1000 μm and 1500 μm from the fovea to nasal and temporal area. (2) CT assessment in the vertical scan: subfoveal and at 500 μm, 1000 μm and 1500 μm from the fovea to superior and inferior area.

### Statistical analysis

A 1 –sample Kolmogorov–Smirnov test was used to verify the normality of data distribution. All tested data were normally distributed, except for sex. Paired Student t-test was used to compare different sectors of CT. The Demographic and clinical participants´ characteristics differences were assessed with Wilcoxon test. The correlation between the OCT findings and the severity of cognitive impairment through MMSE was analyzed by Pearson correlation coefficients.

As both eyes from some patients were included in this study, a univariate generalized estimating equations model (GEE) was used to adjust for these within-patient inter-eye correlations [40, 41]. It was non-viable to include both eyes in some cases due to dementia stage (both eyes assessment was hampered by severe dementia symptoms). Intraclass correlation coefficient (ICC) was used to determine the interobserver reproducibility of manually quantified measurements of CT.

A receiver operating characteristic curve was used to assess the discrimination value of the OCT analyses. We used the area under the receiver operating characteristic curves (AUCs) to assess the ability of CT to discriminate AD/MCI from HC [42].

All statistical analyses were performed using IBM SPSS Statistics V.20.0 (International Business Machine Corporation, Armonk, NY, USA). The level of statistical significance was set at p value less than 0.05.

### Results

Overall, 51 MCI eyes (80.9%) and 12 dementia (19.0%) eyes from 34 patients (from 28 and 6 patients respectively) and 63 eyes from 32 HC were consecutively evaluated in the final analysis based on inclusion and exclusion criteria.

Demographic and clinical characteristics of patients and controls are summarized in Table 1. There were no significant differences among the two groups regarding age, sex, best-corrected visual acuity, intraocular pressure, and axial length measurements. Mean age was 73.1 ± 6.0 years (age range: 57–85 years). All eyes included were phakic.

Table 2 shows the comparison of CT analysis between PiB+ patients and control eyes. Firstly, MCI and dementia patients were compared altogether with HC. CT showed significant reduction across PiB+ compared to HC in vertical scans in foveal thickness (244.9 ± 87.3 µm in PiB+ and 279.9 ± 113.5 µm in HC, p = 0.040), superior scan at $S_{500µm}$ (230.5 ± 85.9 µm in PiB+ and 267.8 ± 116.8 µm in HC, p = 0.026) and inferior scan at $I_{1000µm}$ (243.5 ± 77.8 µm in PiB+ and 281.8 ± 108.9 µm in HC, p = 0.020) and at $I_{1500µm}$ (239.7 ± 76.9 µm in PiB+ and 273.4 ± 109.7 µm in HC, p = 0.045). We found a significant thinning of the CT in horizontal

**Table 1. Demographic and clinical participant's characteristics of Alzheimer disease eyes and control eyes (126 eyes of 66 individuals).**

|  | Patients[a] (N = 63) | Controls (N = 63) | P | MCI (N = 51) | Dementia (N = 12) |
|---|---|---|---|---|---|
| Age (years) | 73.5 (6.0) | 73.28 (6.0) | 0.998 | 73.2 (6.2) | 73.6 (5.5) |
| Male eyes (%) | 31 (49.2) | 31 (49.2) | 1 | 27 (52.9) | 4 (33.3) |
| Spherical equivalent (Diopters) | 0.53 (1.10) | 0.58 (1.22) | 0.797 | 0.61 (1.22) | 0.41 (0.62) |
| BCVA | 20/29 (0.34) | 20/26 (0.17) | 0.259 | 20/28 (0.35) | 20/33 (0.32) |
| Axial length (mm) | 23.2 (0.8) | 23.2 (0.9) | 0.816 | 23.2 (0.8) | 22.9 (0.7) |
| IOP | 13.7 (3.9) | 12.8 (2.8) | 0.154 | 14.0 (4.1) | 12.5 (3.2) |

BCVA, best corrected visual acuity; IOP, intraocular pressure; MCI, mild cognitive impairment; AD, Alzheimer disease.

[a] Patients means MCI and dementia patients due to AD altogether.

Data for quantitative variables are shown as mean (standard deviation). Sex differences were assessed with Fisher's test. Rest of analysis was performed using Wilcoxon test.

**Table 2. Comparison of choroidal thickness analysis between Alzheimer PiB+ patients altogether and control eyes, mild cognitive impairment patients and control eyes and Alzheimer disease patients and controls.**

| | Patients[a] (n = 63) | HCs (n = 63) | P | MCI (n = 51) | HCs (n = 51) | P | Dementia (n = 12) | HCs (n = 12) | P* |
|---|---|---|---|---|---|---|---|---|---|
| CT T 15000µm | 235.4 (18.6) | 266.9 (100.3) | 0.037* | 240.5 (75.0) | 263.6 (85.6) | 0.151 | 213.5 (92.5) | 280.6 (152.0) | 0.104 |
| CT T 1000µm | 242.8 (83.9) | 272.7 (109.1) | 0.056 | 248.4 (81.6) | 270.7 (93.2) | 0.178 | 218.9 (93.0) | 281.5 (165.8) | 0.164 |
| CT T 500µm | 250.5 (88.1) | 287.4 (140.2) | 0.058 | 253.8 (85.8) | 285.6 (130.9) | 0.145 | 236.5 (99.7) | 295.1 (181.3) | 0.195 |
| F H | 253.7 (88.7) | 279.7 (115.0) | 0.121 | 256.8 (87.6) | 277.7 (95.1) | 0.233 | 241.2 (97.4) | 288.2 (182.8) | 0.333 |
| CT N 500µm | 241.1 (88.6) | 267.7 (117.9) | 0.124 | 244.0 (87.9) | 265.0 (98.4) | 0.254 | 228.5 (94.2) | 279.1 (184.8) | 0.301 |
| CT N 1000µm | 225.9 (90.8) | 256.3 (116.9) | 0.088 | 231.3 (91.6) | 254.0 (96.9) | 0.227 | 202.8 (87.1) | 265.8 (185.0) | 0.218 |
| CT N 1500µm | 200.9 (88.6) | 229.23 (113.7) | 0.105 | 206.5 (90.9) | 227.8 (98.4) | 0.253 | 176.9 (77.3) | 235.3 (169.6) | 0.232. |
| CT S 1500µm | 217.0 (84.4) | 240.1 (102.4) | 0.132 | 221.6 (80.5) | 239.6 (90.4) | 0.292 | 197.9 (100.7) | 242.3 (146.5) | 0.237 |
| CT S 1000µm | 226.9 (87.1) | 252.7 (112.9) | 0.123 | 232.2 (83.9) | 251.4 (97.4) | 0.294 | 205.3 (100.5) | 257.8 (167.8) | 0.215 |
| CT S 500µm | 230.5 (85.9) | 267.8 (116.8) | 0.026* | 234.2 (83.7) | 268.41 (101.9) | 0.058 | 215.5 (97.1) | 265.5 (170.8) | 0.271 |
| F V | 244.9 (87.3) | 279.9 (113.5) | 0.040* | 250.5 (86.0) | 279.4 (98.8) | 0.116 | 222.1 (92.7) | 282.3 (163.3) | 0.193 |
| CT I 500µm | 249.2 (84.6) | 278.4 (111.9) | 0.083 | 252.1 (82.6) | 278.5 (99.7) | 0.156 | 237.7 (95.2) | 278.1 (157.6) | 0.331 |
| CT I 1000µm | 243.5 (77.8) | 281.8 (108.9) | 0.020* | 245.8 (74.2) | 283.2 (95.8) | 0.040* | 233.8 (93.9) | 276.0 (155.0) | 0.304 |
| CT I 1500µm | 239.7 (76.9) | 273.4 (109.7) | 0.045* | 243.6 (75.3) | 275.1 (96.9) | 0.083 | 223.8 (84.7) | 266.6 (156.9) | 0.348 |

CT, choroidal thickness; MCI, mild cognitive impairment; AD, Alzheimer disease; HCs, healthy controls; T, temporal; N, nasal; S, superior; I, inferior; FH, subfoveal CT in horizontal scan; FV, subfoveal CT in vertical scan.

[a] Patients means MCI and dementia patients due to AD altogether.

Data for quantitative variables are shown as mean (standard deviation). Analysis was performed using paired Student's t-test for dependent samples.

* p value < 0.05.

scans only in temporal section at $I_{1500\mu m}$ (235.4 ± 18.6 µm in PiB+ and 266.9 ± 100.3 µm in HC, p = 0.037), while we did not find any significant reduction in nasal locations. Secondly, we analyzed CT differences between MCI and HC, finding a CT significant reduction just in $I_{1000\mu m}$ scan (245.8 ± 74.2 µm in MCI and 283.2 ± 95.8 µm in HC, p = 0.040). No significant differences in CT measurements were found in foveal, nasal, superior and inferior scans.

GEE was performed to compare MCI and dementia groups individually with all HC included, shown in Table 3. We found significant thinning of CT between MCI and all HC included in $I_{1000\mu m}$ scan (245.8 ± 74.2 µm in MCI and 281.8 ± 108.9 µm in HC, p = 0.046). Comparing dementia patients with all HC, we found significant thinning in CT $T_{1000\mu m}$ (218.9 ± 93.0 in dementia and 272.7 ± 109.1 in HC, p = 0.046), $S_{1000\mu m}$ (205.3 ± 100.5 in dementia and 252.7 ± 112.9 in HC, p = 0.034), $S_{500\mu m}$ (215.5 ± 97.2 in dementia and 267.8 ± 116.8 in HC, p = 0.034), FV (222.1 ± 92.7 in dementia and 279.9 ± 113.5 in HC, p = 0.037), $I_{1000\mu m}$ (233.8 ± 93.9 in dementia and 281.8 ± 108.9 in HC p = 0.015) and $I_{1500\mu m}$ (223.8 ± 84.7 in dementia and 273.4 ± 109.7 in HC, p = 0.045). GEE test was also used to compare CT measurements between dementia and MCI patients, also in Table 3; no significant thickness reduction was achieved. However, the lack of significant differences among dementia and MCI did not avoid to appreciate that CT results at each location may tend to be thinner in MCI than in HC subjects, and these differences increased in dementia stage.

The AUC analysis was calculated for two different CT measurements, CT $I_{1500\mu m}$ and CT $I_{1000\mu m}$, which were statistically significant, with 95% confidence limits for sensitivity and specificity, as shown in Fig 3. The highest AUC value to discriminate MCI and dementia from HC was CT $I_{1000\mu m}$ (area 0.597, p = 0.062), whereas for $I_{1500\mu m}$ the area was 0.580, p = 0.122. Fig 4 shows the values of $I_{1000\mu m}$ CT in HC, MCI and AD; there is significant difference in CT between HC and MCI or dementia, whose CT values are similar. A trend in CT thinning from HC to MCI and dementia is appreciated in Fig 4.

**Table 3. Comparison of choroidal thickness analysis between mild cognitive impairment and dementia patients due to Alzheimer disease using a univariate generalized estimating equations model.**

| | MCI (n = 51) | AD (n = 12) | P | MCI (n = 51) | HCs (n = 63) | P | Dementia (n = 12) | HCs (n = 63) | P* |
|---|---|---|---|---|---|---|---|---|---|
| CT T $_{1500\mu m}$ | 240.5 (75.0) | 213.5 (92.5) | 0.362 | 240.5 (75.0) | 266.9 (100.3) | 0.125 | 213.5 (92.5) | 266.9 (100.3) | 0.104 |
| CT T $_{1000\mu m}$ | 248.4 (81.6) | 218.9 (93.0) | 0.327 | 248.4 (81.6) | 272.7 (109.1) | 0.165 | 218.9 (93.0) | 272.7 (109.1) | 0.046* |
| CT T $_{500\mu m}$ | 253.8 (85.8) | 236.5 (100.0) | 0.587 | 253.8 (85.8) | 287.4 (140.2) | 0.134 | 236.5 (99.7) | 287.4 (140.2) | 0.103 |
| F H | 256.7 (87.3) | 241.2 (97.4) | 0.619 | 256.7 (87.3) | 279.7 (115.0) | 0.215 | 241.2 (97.4) | 279.7 (115.0) | 0.125 |
| CT N $_{500\mu m}$ | 244.0 (87.9) | 228.5 (94.2) | 0.610 | 244.0 (87.9) | 267.7 (117.9) | 0.234 | 228.5 (94.2) | 267.7 (117.9) | 0.056 |
| CT N $_{1000\mu m}$ | 231.3 (91.6) | 202.9 (87.1) | 0.327 | 231.3 (91.6) | 256.3 (116.9) | 0.210 | 202.8 (87.1) | 256.3 (116.9) | 0.171 |
| CT N $_{1500\mu m}$ | 206.5 (90.0) | 176.9 (77.3) | 0.264 | 206.5 (90.0) | 229.23 (113.7) | 0.218 | 176.9 (77.3) | 229.23 (113.7) | 0.125 |
| CT S $_{1500\mu m}$ | 221.6 (80.5) | 197.9 (101.0) | 0.460 | 221.6 (80.5) | 240.1 (102.4) | 0.271 | 197.9 (100.7) | 240.1 (102.4) | 0.105 |
| CT S $_{1000\mu m}$ | 232.2 (83.9) | 205.3 (100.5) | 0.406 | 232.2 (83.9) | 252.7 (112.9) | 0.278 | 205.3 (100.5) | 252.7 (112.9) | 0.034* |
| CT S $_{500\mu m}$ | 234.2 (83.7) | 215.5 (97.2) | 0.548 | 234.2 (83.7) | 267.8 (116.8) | 0.105 | 215.5 (97.1) | 267.8 (116.8) | 0.034* |
| F V | 250.5 (86.0) | 222.1 (92.7) | 0.348 | 250.5 (86.0) | 279.9 (113.5) | 0.098 | 222.1 (92.7) | 279.9 (113.5) | 0.037* |
| CT I $_{500\mu m}$ | 252.1 (82.6) | 237.7 (95.2) | 0.637 | 252.1 (82.6) | 278.4 (111.9) | 0.123 | 237.7 (95.2) | 278.4 (111.9) | 0.121 |
| CT I $_{1000\mu m}$ | 245.9 (74.2) | 233.8 (93.9) | 0.684 | 245.8 (74.2) | 281.8 (108.9) | 0.046* | 233.8 (93.9) | 281.8 (108.9) | 0.015* |
| CT I $_{1500\mu m}$ | 243.6 (75.3) | 223.8 (84.7) | 0.470 | 243.6 (75.3) | 273.4 (109.7) | 0.062 | 223.8 (84.7) | 273.4 (109.7) | 0.045* |

CT, choroidal thickness; MCI, mild cognitive impairment; AD, Alzheimer disease; HCs, healthy controls; T, temporal; N, nasal; S, superior; I, inferior; FH, subfoveal CT in horizontal scan; FV, subfoveal CT in vertical scan.

Data for quantitative variables are shown as mean (standard deviation).

*p value < 0.05.

As CT is a subjective measurement, we calculated the intraclass correlation coefficient (ICC), used to determine the interobserver reproducibility of manually quantified measurements in Table 4, showing an excellent reliability [43].

Table 5 shows the correlation between MMSE score and CT at each location. Despite a non-significant and very weak correlation between the parameters (p>0.141), a tendency of CT thinning in association with MMSE decreased was observed in all measurements and we considered it a consistent positive correlation coefficient.

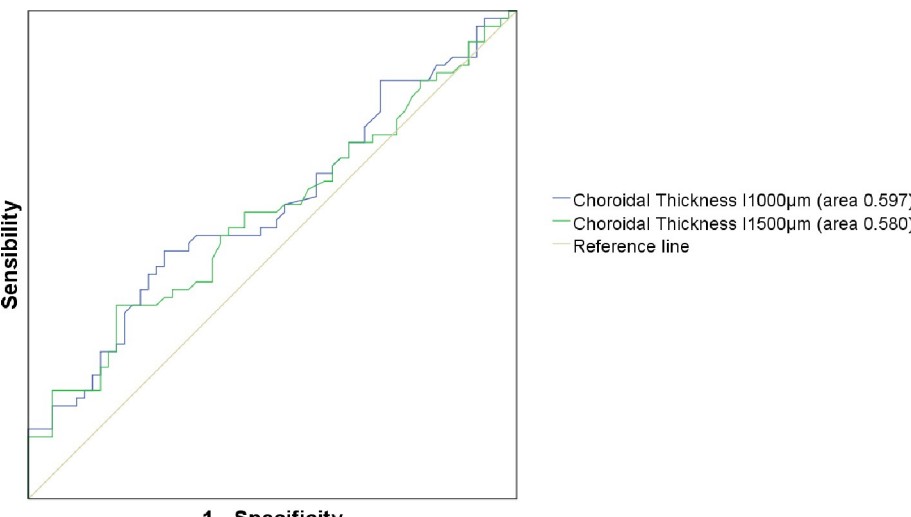

**Fig 3. AUC of CT I$_{1500\mu m}$ and CT I$_{1000\mu m}$.** The area under the curve (AUC) of choroidal thickness (CT) at 1000 μm in inferior (I) (blue line) and at 1500 μm in inferior (green line) had the highest values under the curve and represented the most sensibility and specificity measurements.

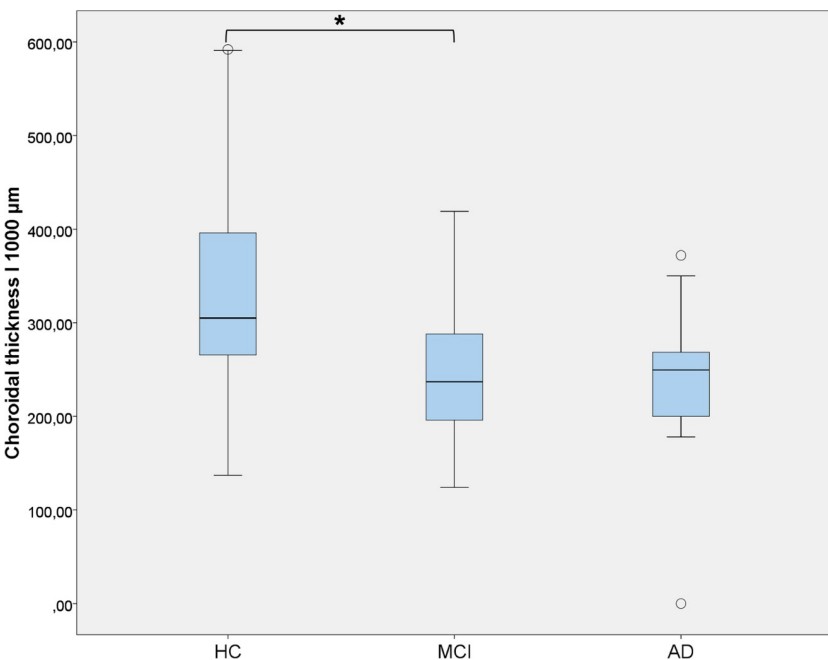

**Fig 4. AUC of CT I$_{1000\mu m}$ in MCI patients, dementia patients and HCs.** Choroidal thickness (CT) at 1000 μm in inferior (I) sector in healthy controls (HCs), mild cognitive impairment (MCI) and dementia due Alzheimer's disease (AD) eyes (error bars represent 95% confidence intervals), described on a bar chart.

## Discussion

Nowadays, the gold standard for diagnosing AD is based on laboratory biomarkers (such as Aβ-42, P-tau, T-tau) and imaging techniques (MRI and PET using amyloid tracers) [6–13].

**Table 4. Intraclass correlation coefficient used to determine interobserver reproducibility of manually quantified measurements and the confidence intervals associated.**

|  | ICC | 95% CI |
|---|---|---|
| CT T $_{1500\mu m}$ | 0.997 | 0.996–0.999 |
| CT T $_{1000\mu m}$ | 0.976 | 0.952–0.985 |
| CT T $_{500\mu m}$ | 0.988 | 0.981–0.994 |
| F$_H$ | 0.980 | 0.973–0.991 |
| CT N $_{500\mu m}$ | 0.979 | 0.966–0.989 |
| CT N $_{1000\mu m}$ | 0.944 | 0.915–0.972 |
| CT N $_{1500\mu m}$ | 0.951 | 0.914–0.972 |
| CT S $_{1500\mu m}$ | 0.957 | 0.930–0.978 |
| CT S $_{1000\mu m}$ | 0.965 | 0.940–0.981 |
| CT S $_{500\mu m}$ | 0.966 | 0.938–0.980 |
| F$_V$ | 0.944 | 0.908–0.971 |
| CT I $_{500\mu m}$ | 0.964 | 0.944–0.982 |
| CT I $_{1000\mu m}$ | 0.916 | 0.857–0.954 |
| CT I $_{1500\mu m}$ | 0.905 | 0.831–0.942 |

ICC, intraclass correlation coefficient; CT, choroidal thickness; T, temporal; N, nasal; S, superior; I, inferior. FH, subfoveal choroidal thickness in horizontal scan; FV, subfoveal choroidal thickness in vertical scan; CI, confidence intervals.

Table 5. Correlation of mini-mental state examination with each measurement of choroidal thickness.

| | r value | p value* |
|---|---|---|
| CT T 1500μm | 0.063 | 0.647 |
| CT T 1000μm | 0.055 | 0.687 |
| CT T 500μm | 0.055 | 0.690 |
| F H | 0.055 | 0.691 |
| CT N 500μm | 0.092 | 0.505 |
| CT N 1000μm | 0.079 | 0.567 |
| CT N 1500μm | 0.103 | 0.453 |
| CT S 1500μm | 0.214 | 0.124 |
| CT S 1000μm | 0.205 | 0.141 |
| CT S 500μm | 0.158 | 0.258 |
| F V | 0.159 | 0.256 |
| CT I 500μm | 0.119 | 0.397 |
| CT I 1000μm | 0.108 | 0.441 |
| CT I 1500μm | 0.140 | 0.317 |

CT, choroidal thickness; T, temporal; N, nasal; S, superior; I, inferior; FH, subfoveal CT in horizontal scan; FV, subfoveal CT in vertical scan.

Analysis was performed using Pearson´s correlation coefficient.

No significant results were achieved regarding CT and Mini-Mental State Examination correlation (*p>0.05)

These widely investigated biomarkers for the molecular and degenerative process of AD can be supportive of AD diagnosis but they are not recommended for routine diagnostic purposes just in clinical trials and research studies [4]. Recently, increasing efforts have been made to discover new biomarkers with the aim to improve AD diagnosis in early stages. In an attempt to investigate CT in AD patients, we conducted a study which involved deeply characterized prodromal AD patients with detailed neurocognitive testing and PET imaging with [11]C-PiB. The present work is a logical extension of our previous publication about AD biomarkers, in which we suggested RNFL and RGCL as potential AD biomarkers in a near future [25]. Our main outcomes herein were CT thinning in different localizations comparing PiB+ patients (MCI stage and dementia stage) versus HC and a general trend toward CT thinning in MCI patients compared with HC, which became more pronounced in dementia. The choroid is a highly vascularized layer that supplies the outer retina with oxygen, nutrients and growth factors. It also serves as a heat diffuser, protecting the photoreceptors [31]. Growing evidence about the choroidal involvement in AD [26–30] and the development of EDI-OCT technology has provided a chance to identify new visual non-invasive biomarkers [34]. Likely related to cerebral vascular impairment in early AD [44, 45], choroidal thinning may represent a novel biomarker of AD.

To the best of our knowledge, the present work constitutes the first study investigating the thickness of choroidal tissue in MCI subjects positive to [11]C-PiB PET/CT. Our findings showed significant differences between CT across PiB+ patients in selected locations (CT $T_{1000\mu m}$, CT $I_{1000\mu m}$ and $_{1500\mu m}$, CT $S_{1500\mu m}$ and foveal thickness in vertical scan) but few statistically significant CT reduction data in MCI or dementia groups individually versus HC. Although our primary finding was the lack of association between CT thinning and MCI or dementia versus age–and sex- matched HC, an interesting observation emerged from a more powerful statistical analysis performed. GEE was performed to compare MCI and dementia patients individually with all HC included, shown in Table 3 and we demonstrated a significant reduction of one choroid measurements in dementia patients. Even more, a general trend

toward the CT thinning in MCI patients, which became more pronounced in dementia, is shown in **Table** 3. This tendency is slightly appreciated in Fig 3. The inclusion of one eye per subject, randomly selected, is widely spread for statistical purposes, reducing possible bias of side preference. Actually, most studies proceed this way [29]. Nevertheless, there are complex statistical analysis which allow the use of both eyes without bias increasing the sample size [46].

Previous data determined that cerebral vascular damage, due to accumulation of A$\beta$ [12], plays an important role in early AD progression [44, 45, 47]. In accordance to this, it has been hypothesized that as A$\beta$ deposition causes angiopathy in the brain, it might cause angiopathy in the choroid; and, subsequently, atrophy of choroidal tissue reflected in a reduction in CT [48, 49]. Supporting this idea, both A$\beta$ plaques and Tau neurofibrillary tangles (NFTs) have been detected in some parts of the visual system in AD patients, including the retina [50, 51]. Interestingly, in a mouse model of AD, A$\beta$ deposits were specifically located in the RGCL [48]. Accordingly, Koronyo et al demonstrated histopathologically that RGCL thinning due to AD might be related with intracellular NFTs of Tau and extracellular A$\beta$ protein deposits throughout the retina and not related with other etiologies of dementia [52]. Similarly, A$\beta$ accumulation has been detected in choroidal tissue in normal aging mice, in several mouse models of AD and in human post-mortem retina samples from AD donors [47–49]. We assumed choroidal thinning might be related to a series of pathologic events triggered by A$\beta$ accumulation.

Several OCT studies showed choroidal thinning using EDI technology in mild and moderate dementia-AD [26–29] and one study showed CT thinning in MCI [30]. Every single study, except one [26], performed EDI-OCT examination and measured the perpendicular CT from the outer edge of the hyperreflective retinal pigment epithelium to the inner sclera, getting similar average thickness of the choroid, in agreement with the work hereby presented. However, a different number of CT locations were measured in each study, within 7 [30], 9 [26–28] and 13 [29] measurements. In order to solve this discrepancy, we analyzed CT at 14 locations, 2 of subfoveal thickness in 2 different scans and 12 more separated 1500, 1000 and 500 μm from these subfoveal locations [37].

Despite the disparity of the analyzed data, all investigations described a significantly thinning of CT measurements at each location, among AD and HC. Bulut et al added significant differences in MCI and Cunha et al showed a significant choroidal thinning in AD versus age matched controls and even when compared with elderly subjects [29, 30]. In addition, Gharabiya´s group took measurements at baseline and 12 months later, reporting CT decreased significantly after this time in the AD group whereas no significant reduction was observed in controls [27].

In our study, we did not find a significant correlation between the CT values at all localizations and the MMSE scores. In agreement with our results, Bayhan et al and Gharbiya et al reported no significant correlation between CT and each of the tested psychometric parameters [26, 27]. Trebastoni et al conducted a prospective study which measured CT at baseline and after one year, describing cognitive functions deterioration assessed by MMSE, Alzheimer's Disease Assessment Scale-Cognitive (ADAS-Cog 11), and Clinical Dementia Rating Scale (CDR) at the end of these 12 months (p<0.0001), but no correlations were found between psychometric scores' changes and neither baseline CT nor CT changes [28]. In contrast, Bulut et al observed a significantly positive correlation between MMSE score and CT value [30]. Although current diagnosis of AD is based on cognitive clinical evaluation, such an approach might be insufficient in individuals with much cognitive reserve and we hypothesize that the lack of correlation in our study could reflect MMSE may not be a suitable test to detect subtle and initials changes at early disease stages like MCI or mild dementia.

As far as we are concerned, the aforementioned studies have three main limitations. Firstly, none of them supported their findings with CSF biomarkers or PET exams to diagnose AD

patients. This implies a variable degree of case misclassification affecting statistical power and the interpretation of the outcome. Hence, the use of AD biomarkers cannot be ignored in the design of OCT studies. We emphasize the importance of enrolling patients based AD biomarkers status instead of MMSE-based criteria. Secondly, those OCT studies faced another important limitation concerning their section. Patients´ eyes assessment underwent a complete ophthalmologic evaluation, including dilated fundus examination. Nevertheless, they did not mention if OCT analysis was performed before or after pupil dilatation nor did specify which kind of drop was used. This may be of crucial importance, as we have proved that phenylephrine 2.5%, a common drop for pupil dilatation, might cause a significant choroidal thinning thirty minutes after its instillation [37]. Thus, mentioning the use of dilatation drops should be important to reach a conclusion in CT changes, because if they used phenylephrine differently in AD patients and controls, this might be a source of bias. Finally, these studies assessed CT using exclusively 7 to 13 locations. As choroidal analysis was based on subjective and non-automated measurements, we analyzed it in 14 locations in order to reduce bias as we previously reported [37].

The main limitations of the present study are the relatively small sample size and the cross-sectional design. Specifically, our study included few patients with dementia due to AD in order to depict if there was trend of CT to be thinner in worse stages of dementia. However, our sample was characterized by a marked homogeneity in ocular biometric parameters that strengthens the power of our results. Even so, future research should include a higher number of subjects with both early and late stage AD and longitudinal measurements. Another common limitation is the measurement of CT manually using EDI-OCT, providing us a choroidal analysis based on subjective, non-automated measurements. To help overcome this hurdle, the study was designed taking this into account, and a well-trained ophthalmologist (AC) unaware of patients' diagnoses performed the CT measurements. Besides, this manual technique had already been used in previous reports [34] and proved to have high intra-observer and inter-observer reproducibility [53].

One of the major advantages of the present work is that the research protocol was undertaken in a real clinical setting in well characterized MCI patients. Hence, our results represent very likely day-to-day in clinical practice.

In conclusion, our study described CT thinning in selected localizations, but not a statistical significant and general choroidal thinning comparing dementia and MCI versus HC. For this reason, CT might be a promising target to find a biomarker in prodromal stages of AD because there is a general choroidal reduction trend from HC to MCI patients, which become slightly more pronounced in AD. It would be interesting to conduct larger and prospective OCT studies in AD population characterized with available biomarkers to describe whether choroidal vascular damage occurs specifically in prodromal stages of AD.

## Author Contributions

**Formal analysis:** Alfonso Casado.

**Investigation:** Alicia López-de-Eguileta, Carmen Lage, Sara López-García, Ana Pozueta, María García-Martínez, Martha Kazimierczak, María Bravo, María de Arcocha-Torres, Ignacio Banzo, Julio Jimenez-Bonilla, Andrea Cerveró, Eloy Rodríguez-Rodríguez, Pascual Sánchez-Juan, Alfonso Casado.

**Methodology:** Alicia López-de-Eguileta, Sara López-García, Pascual Sánchez-Juan, Alfonso Casado.

**Project administration:** Alicia López-de-Eguileta, Pascual Sánchez-Juan.

**Supervision:** Sara López-García, Alexander Goikoetxea, Alfonso Casado.

**Validation:** Pascual Sánchez-Juan.

**Writing – original draft:** Alicia López-de-Eguileta, Alfonso Casado.

**Writing – review & editing:** Alicia López-de-Eguileta, Andrea Cerveró, Alexander Goikoetxea, Pascual Sánchez-Juan, Alfonso Casado.

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
