## [Decision Letter · Decision Letter 0]

1 Jul 2020

PONE-D-20-16899

Evaluation of choroidal thickness in prodromal Alzheimer´s disease defined by amyloid PET

PLOS ONE

Dear Dr. López de Eguileta,

Thank you for submitting your manuscript to PLOS ONE. After careful consideration, we feel that it has merit but does not fully meet PLOS ONE’s publication criteria as it currently stands. Therefore, we invite you to submit a revised version of the manuscript that addresses the points raised during the review process.

The reviewer found the article interesting and made several suggestions improve it. I agree with hem and we look forward to the revised version

We look forward to receiving your revised manuscript.

Kind regards,

Demetrios G. Vavvas

Academic Editor

PLOS ONE

Journal Requirements:

2. We understand that the submitted manuscript may be closely related to a previous publication, authored by you/your co-authors:

https://doi.org/10.1016/j.trci.2019.08.008

As we note on our website, upon submission of a manuscript, authors must indicate whether there are any related manuscripts under consideration or published elsewhere. If related work has been submitted or published elsewhere, authors must include a copy of it with their submission and describe its relation to the submitted work. (https://journals.plos.org/plosone/s/ethical-publishing-practice#loc-submission-and-publication-of-related-studies). Please revise your submission to respond to the following points:

A) If the study cohort of the present submission overlaps with your previous work this should clearly be indicated in the Methods section.

B) If any of the results (including basic demographic information) have been reported elsewhere this should be clearly indicated in the Results section.

C) Please discuss how the present work advances on your previous publication (referenced above) in the Discussion section.

3. Please describe in your methods section how capacity to provide consent was determined for the participants in this study. Please also state whether your ethics committee or IRB approved this consent procedure. If you did not assess capacity to consent please briefly outline why this was not necessary in this case.

4. Thank you for stating the following in the Title page of your manuscript:

"Dr. Sánchez-Juan was supported by grants from IDIVAL, Instituto de Salud Carlos III (Fondo de Investigación Sanitario, PI08/0139, PI12/02288, PI16/01652, JPND (DEMTEST PI11/03028) and the CIBERNED program and Siemens Healthineers (Valdecilla Cohort for Memory and Brain Aging)..."

a. Please remove any funding-related text from the manuscript and let us know how you would like to update your Funding Statement.

Currently, your Funding Statement reads as follows:

i. Please clarify the sources of funding (financial or material support) for your study. List the grants or organizations that supported your study, including funding received from your institution.

ii. State what role the funders took in the study. If the funders had no role in your study, please state: “The funders had no role in study design, data collection and analysis, decision to publish, or preparation of the manuscript.”

iii. If any authors received a salary from any of your funders, please state which authors and which funders.                                                                                                                                            

If you did not receive any funding for this study, please state: “The authors received no specific funding for this work.”

b. Additionally, because some of your funding information pertains to commercial funding, we ask you to provide an updated Competing Interests statement, declaring all sources of commercial funding.

In your Competing Interests statement, please confirm that your commercial funding does not alter your adherence to PLOS ONE Editorial policies and criteria by including the following statement: "This does not alter our adherence to PLOS ONE policies on sharing data and materials.” as detailed online in our guide for authors http://journals.plos.org/plosone/s/competing-interests. If this statement is not true and your adherence to PLOS policies on sharing data and materials is altered, please explain how.

c. Please include the updated Competing Interests Statement and Funding Statement in your cover letter. We will change the online submission form on your behalf.

Please know it is PLOS ONE policy for corresponding authors to declare, on behalf of all authors, all potential competing interests for the purposes of transparency. PLOS defines a competing interest as anything that interferes with, or could reasonably be perceived as interfering with, the full and objective presentation, peer review, editorial decision-making, or publication of research or non-research articles submitted to one of the journals. Competing interests can be financial or non-financial, professional, or personal. Competing interests can arise in relationship to an organization or another person. Please follow this link to our website for more details on competing interests: http://journals.plos.org/plosone/s/competing-interests'

5. Please amend your list of authors on the manuscript to ensure that each author is linked to an affiliation. Authors’ affiliations should reflect the institution where the work was done (if authors moved subsequently, you can also list the new affiliation stating “current affiliation:….” as necessary).

Reviewers' comments:

Reviewer's Responses to Questions

**Comments to the Author**

1. Is the manuscript technically sound, and do the data support the conclusions?

Reviewer #1: Yes

2. Has the statistical analysis been performed appropriately and rigorously? 

Reviewer #1: N/A

3. Have the authors made all data underlying the findings in their manuscript fully available?

Reviewer #1: Yes

4. Is the manuscript presented in an intelligible fashion and written in standard English?

Reviewer #1: Yes

5. Review Comments to the Author

Reviewer #1: Thank you for submitting your research work to PLOS ONE.

This is an interesting study but I would like to make some comments and suggestions.

Abstract

Line 81: this is the first time we see GEE so do not use its abbreviation here

Line 86: which are these locations?

Try to present your results with % here and throughout the manuscript.

Introduction

I believe that this part could be improved. More specifically, you could dedicate a paragraph in order to talk more about Alzheimer’s disease including also recent epidemiological data. Then, in your second paragraph you could include the diagnostic approaches (also lines 109 to 125). The third paragraph could be the one that starts in line 126. However, I would suggest you add 1-2 sentences talking more about eyes-brain connection and why we could use OCT findings in order to draw conclusions for CNS processes.

In general, the findings of some of the studies that you have included here could be instead added to the discussion section.

Line 106: you may want to say “stage” instead of “state”.

Line 150: anatomical variations

Methods

Frankly, I do not believe that anyone (especially a medical doctor) who interacts with patients can be totally blinded to the neurological status of a patient to some degree at least.

In addition, is there any particular reason why all CT measurements were taken between 5pm and 6pm? If yes, please cite accordingly.

Line 178: end the sentence with “.”

It would be useful to add representative images of positive 11C-PiB PET/CT and age-matched healthy controls or of patients who fulfilled clinical criteria of AD but tested negative.

Please, comment on the specificity and sensitivity of 11C-PiB PET/CT.

Statistical analysis

So, did you have normally distributed data or not? Based on the choice of all tests it is quite confusing. Since you include both eyes you could use multilevel mixed-effect linear models.

Line 210: chorioretinopathy not corioretinopathy

Results

I believe that it is not necessary to include all these tables in the main manuscript.

Discussion

You have included many interesting findings from other studies. However, similar to the Introduction section you could probably improve the flow.

Please, talk about the gold standard for diagnosing Alzheimer’s disease and comment on your choice to use non validated biomarkers in order to show that other tests could be promising biomarkers.

Line 426: To the best of our knowledge . . . .

Also, after this phrase you should actually make a more confident statement.

Line 493: as far as we are concerned

Lines 502, 506: use the word “mention” instead of “detail”

Line 525: do you mean “major advantages”?

Looking forward to your revised manuscript.

Thank you.

6. PLOS authors have the option to publish the peer review history of their article (what does this mean?). If published, this will include your full peer review and any attached files.

Reviewer #1: No

---

## [Author Response · Author response to Decision Letter 0]

21 Aug 2020

PONE-D-20-16899R1

Evaluation of choroidal thickness in prodromal Alzheimer´s disease defined by amyloid PET

Mrs Alicia López de Eguileta

Dear Dr. López de Eguileta,

We've checked your submission and before we can proceed, we need you to address the following issues:

1..Thank you for responding to our previous regarding your manuscript. We note that the participants included patients with dementia. Please clarify how patients with dementia were determined capable of providing informed consent. If you did not assess the capacity to provide consent please outline why this was not necessary in this case. Thank you for your attention to this request.

As you suggested, to clarified how patients with dementia were determined capable of providing informed consent, we added this in Methods section in the manuscript: ”All patients enrolled were able to understand the information contained in the written consent and they were not legally incompetent.“

2..Based on the information you have provided, we propose the following funding and competing interests for your approval:

Funding:

Dr. Sánchez-Juan was supported by grants from IDIVAL, Instituto de Salud Carlos III (Fondo de Investigación Sanitario, PI08/0139, PI12/02288, PI16/01652, JPND (DEMTEST PI11/03028) and the CIBERNED program and Siemens Healthineers (Valdecilla Cohort for Memory and Brain Aging). The funders had no role in study design, data collection and analysis, decision to publish, or preparation of the manuscript.

Competing interests:

Dr. Sánchez-Juan received funding from Siemens Healthineers. This does not alter our adherence to PLOS ONE policies on sharing data and materials.

Can you please confirm in your cover letter the above statements are both complete and correct? With your approval we will update your competing interests and funding statements on your behalf.

We confirmed the above statements are both complete and correct and we added them in the Cover letter.

---

## [Decision Letter · Decision Letter 1]

8 Sep 2020

Evaluation of choroidal thickness in prodromal Alzheimer´s disease defined by amyloid PET

PONE-D-20-16899R1

Dear Dr. López de Eguileta,

We’re pleased to inform you that your manuscript has been judged scientifically suitable for publication and will be formally accepted for publication once it meets all outstanding technical requirements.

Kind regards,

Demetrios G. Vavvas

Academic Editor

PLOS ONE

Additional Editor Comments (optional):

Reviewers' comments:

Reviewer's Responses to Questions

**Comments to the Author**

1. If the authors have adequately addressed your comments raised in a previous round of review and you feel that this manuscript is now acceptable for publication, you may indicate that here to bypass the “Comments to the Author” section, enter your conflict of interest statement in the “Confidential to Editor” section, and submit your "Accept" recommendation.

Reviewer #1: All comments have been addressed

Reviewer #2: All comments have been addressed

2. Is the manuscript technically sound, and do the data support the conclusions?

Reviewer #1: Partly

Reviewer #2: Yes

3. Has the statistical analysis been performed appropriately and rigorously? 

Reviewer #1: N/A

Reviewer #2: Yes

4. Have the authors made all data underlying the findings in their manuscript fully available?

Reviewer #1: Yes

Reviewer #2: Yes

5. Is the manuscript presented in an intelligible fashion and written in standard English?

Reviewer #1: Yes

Reviewer #2: Yes

6. Review Comments to the Author

Reviewer #1: I would like to thank you for taking the time to address these issues.

Looking forward to reading the revised manuscript.

Reviewer #2: The authors have addressed all previous comments in an adequate manner and their work has been substantially improved - no further comments on my end.

7. PLOS authors have the option to publish the peer review history of their article (what does this mean?). If published, this will include your full peer review and any attached files.

Reviewer #1: No

Reviewer #2: No

---

## [Editor Report · Acceptance letter]

11 Sep 2020

PONE-D-20-16899R1 

Evaluation of choroidal thickness in prodromal Alzheimer´s disease defined by amyloid PET 

Dear Dr. López-de-Eguileta:

I'm pleased to inform you that your manuscript has been deemed suitable for publication in PLOS ONE. Congratulations! Your manuscript is now with our production department. 

Kind regards, 

on behalf of

Dr. Demetrios G. Vavvas 

Academic Editor

PLOS ONE